

# Biochar improves the growth and physiological traits of alfalfa, amaranth and maize grown under salt stress

Dilfuza Jabborova[1,2], Tokhtasin Abdrakhmanov[1], Zafarjon Jabbarov[1], Shokhrukh Abdullaev[1], Abdulahat Azimov[2], Ibrahim Mohamed[3], Maha AlHarbi[4], Abdelghafar Abu-Elsaoud[5,6] and Amr Elkelish[5,6]

[1] National University of Uzbekistan, Tashkent, Uzbekistan
[2] Uzbekistan Academy of Sciences, Kibray, Uzbekistan
[3] Benha University, Benha, Egypt
[4] Princess Nourah bint Abdulrahman University, Riyadh, Saudi Arabia
[5] Suez Canal University, Ismailia, Egypt
[6] Imam Mohammad ibn Saud Islamic University, Riyadh, Saudia Arabia

Corresponding authors
Dilfuza Jabborova,
dilfuzajabborova@yahoo.com
Amr Elkelish,
amr.elkelish@science.suez.edu.eg

## ABSTRACT

**Purpose:** Salinity is a main factor in decreasing seed germination, plant growth and yield. Salinity stress is a major problem for economic crops, as it can reduce crop yields and quality. Salinity stress occurs when the soil or water in which a crop is grown has a high salt content. Biochar improve plant growth and physiological traits under salt stress. The aim of the present study, the impact of biochar on growth, root morphological traits and physiological properties of alfalfa, amaranth and maize and soil enzyme activities under saline sands.

**Methods:** We studied the impact of biochar on plant growth and the physiological properties of alfalfa, amaranth and maize under salt stress conditions. After 40 days, plant growth parameters (plant height, shoot and root fresh weights), root morphological traits and physiological properties were measured. Soil nutrients such as the P, K and total N contents in soil and soil enzyme activities were analyzed.

**Results:** The results showed that the maize, alfalfa, and amaranth under biochar treatments significantly enhanced the plant height and root morphological traits over the control. The biochar on significantly increased the total root length, root diameter, and root volume. Compared to the control, the biochar significantly increased the chlorophyll a and b content, total chlorophyll and carotenoid content under salt stress. Furthermore, the biochar significantly increased enzyme activities of soil under salt stress in the three crops.

**Conclusions:** Biochar treatments promote plant growth and physiological traits of alfalfa, amaranth, and maize under the salt stress condition. Overall, biochar is an effective way to mitigate salinity stress in crops. It can help to reduce the amount of salt in the soil, improve the soil structure, and increase the availability of essential nutrients, which can all help to improve crop yields.

## INTRODUCTION

Soil degradation in the Republic of Uzbekistan is mainly caused by salinity; 54% of the irrigated lands of Uzbekistan are subject to varying degrees of salinity. This salinity is especially true in the Republic of Karakalpakistan, Khorezm, Bukhara, Syrdarya and Jizzakh regions (*Qi & Evered, 2008*). Among soil degradation in the world, soil salinity plays a major role; the occurrence of salinity affects the amount of carbon (C), nitrogen (N), carbon emission, bacterial gene copy number, actinobacteria, thermophilic and beta proteobacteria in soil (*Yang et al., 2021*). Soil salinity is related to soil physical properties; the effects of wind erosion on salts in soil were studied. These studies were conducted on samples treated with salts in an arid region, and wind tunnel tests and a strong wind of 18 m/c were conducted under the influence; according to the results, $Na_2SO_4$, $MgSO_4$ and $Na_2CO_3$ formed high emissive surfaces, $Na_2SO_4$ and $Na_2CO_3$ crystals appear in sharp form, dehydration fine aggregates for $MgSO_4$ were the primary source of dust (*Dai et al., 2022*). Soil salinization negatively affects soil processes, including carbon-nitrogen ratio, phosphatase, cellulase enzyme activity, and microbiological diversity (*Yang et al., 2020*).

Salinity harms the nutrition of plants in the soil; the use of rhizobacteria with compost in such soils reduces the amount of sodium (Na+) in the soil, and affects the activity of soil enzymes resulting in the improvement of plant nutrition (*Omara et al., 2022*). *Van Breusegem et al. (2001)* reported that biochemical changes occurring when plants are subjected to salt stress is the accumulation of reactive oxygen species (superoxide and hydrogen peroxide). According to earlier studies, enzymatic antioxidants such as superoxide dismutase, catalase, peroxidase and ascorbate peroxidase have been reported antioxidant enzymes in salinity stress (*Abdel Latef & Chaoxing, 2014*; *Evelin et al., 2019*). *Kahrizi, Sedghi & Sofalian (2012)* reported that salinity stress enhanced peroxidase activity in wheat cultivars.

The drying up of the Aral Sea was caused by global climate change and the irregular use of water resources since 1960, the poor functioning of drainage collectors and the expansion of irrigated agriculture in the region, resulting in a decline in the water level and an enhance in salinity (*Micklin, 1988*). Also, the transition from hydromorphic to automorphic accelerated, and desertification became more active. The Aral Sea's drying up, and the soil formation process began in the open sand dunes (*Assouline et al., 2015*). Aral Sea, the amount of water-soluble salts in the soil increased from 4–5 to 71.3 g/L; according to studies, the number of microorganisms decreased with the increase of salts in the soil (*Jiang et al., 2021*). A decrease in plant life has been observed around the South Aral Sea. Therefore, monitoring and increasing plant life ensure ecological stability (*Kochkarova, 2020*). The increase of plants in the dry bottom of the island leads to the improvement of soil properties; in this regard, the concentration of cations in the soil ($Ca^{2+}$, $K^+$, $Mg^{2+}$ va $Na^+$), cation exchange capacity, pH environment, enzyme activity (phosphatase, b-glucodase, and N-acetylglucosaminidase) in the 0–10 cm layer, when planting plants decreased the concentration of basic cations, and electrical conductivity, the activity of enzymes and the number of microorganisms have increased (*An et al., 2020*), the widespread introduction of phytoremediation on the island prevents sand flying

(*Issanova et al., 2015*). In the dry regions of the Aral Sea, the salinity type of soils is chloride-sulfate; the amount of salts is from 2.09% to 4.21%, the amount of chlorine is from 0.59% to 0.82%, sulfate is from 0.68% to 2.24%, sodium is 0.67% from 1.08%, black saxovol was planted in the studies, its phytomass (58.7 t/ha), 669 16-year-old seedlings per 1 ha, phytomass 185.9 t/ha, 1,682 saxovol per hectare the seedling is correct (*Duan, Wang & Sun, 2022*).

Aral Sea, where the study was conducted, does not have complete soil formation at the moment because if we take into account the participation of six factors in the formation of soil and assume that 1 cm of soil is formed on average in 200 years if it is considered that it has been 60 years since the drying of the island until now, there is still a 1 cm layer in the dry bottom of the island. There is no soil, and the process of soil formation has started in some areas (*Jabborova et al., 2022*). From this point of view, it can be said that scientists and experts who plan to conduct scientific, practical and innovative research in the dry bottom of the Aral Sea should take into account the absence of soil cover.

Biochar application to soil plays an essential role in promoting nutrient dynamics and modifying soil pollutants and microbial functions (*Shaaban et al., 2018*). Biochar continues to be used in agriculture, promoting soil properties and improving productivity (*Day et al., 2004*). The effect of biochar is either directly by supplying nutrients to plants or indirectly by improving soil properties, resulting in increased plant nutrient utilization efficiency (*Keeney, 2015*). Biochar improves soil quality and crop productivity; its effect is realized through greater soil microbial mass, enzyme activity, including microbial biomass depending on soil types, urease, alkaline phosphatase, and dehydrogenase enzyme activity, respectively 21.7%, 23.1%, 25.4% increased to 19.8% (*Pokharel, Ma & Chang, 2020*), 5 years after the application of biochar, the carbon in the soil increased by 2.44%, and the amount of oxygen increased by 2.81% (*Dong et al., 2017*). Biochar affects the changes in the physical properties of soils, including improved filtration and moisture retention properties, enhanced $CO_2$, $N_2O$ and $CH_4$ gas formation, reduced granulometric composition when applied to heavy soils, reduced density, increased porosity, increased plant water use (*Kinney et al., 2012*; *Masiello et al., 2015*).

Several studies have shown that salt stress decreased seed germination (*Ashraf & McNeilly, 2004*) and plant growth (*Batool et al., 2015*; *Jabborova et al., 2020*; *Jabborova et al., 2021a*, *2021b*; *Kumar et al., 2021a*; *Menezes et al., 2017*). Moreover, on the root morphological traits (*Wang et al., 2008*; *Zeeshan et al., 2020*) and physiological properties (*Azari et al., 2012*; *Jabborova et al., 2021d*, *2021e*; *Saddiq et al., 2021*) in plants. *Akhtar, Andersen & Liu (2015)* reported that growth, physiology and yield of wheat by affect positively with biochar amendment under salt stress. *Zulfiqar et al. (2021)* informed that biochar enhance chlorophyll a and b contents and net photosynthetic rates of *Alpinia zerumbet* compared with those grown in the sandy loam soil. *Mehdizadeh, Moghaddam & Lakzian (2020)* reported that biochar increase the growth, physiological traits and mineral nutrient contents of summer savory (*Satureja hortensis* L.) and under salinity condition.

Biochar's beneficial impact on plant nutrients (*Jabborova et al., 2022*, *2021c*) and enzyme activities in soil (*Jabborova et al., 2021e*; *Song et al., 2022*) were investigated in various plants under normal and stress conditions. *Nikpour-Rashidabad et al. (2019)*

**Table 1 The municipal solid waste biochar characteristics.**

| Biochar | BOC (g/kg) | BOM (g/kg) | TN (g/kg) | TP (g/kg) | TK (g/kg) | AN (mg/kg) | AP (mg/kg) | AK (mg/kg) | pH |
|---|---|---|---|---|---|---|---|---|---|
| Mean contents | 330.6 | 570.0 | 0.26 | 4.48 | 42.6 | 237.8 | 0.77 | 688.9 | 8.01 |

Note:
BOC, biochar organic carbon (g/kg); BOM, biochar organic matter (g/kg); TN, total nitrogen (g/kg); TP, total phosphorous (g/kg); TK, total potassium (g/kg); AN, available nitrogen (mg/kg); AP, available phosphorous (mg/kg); AK, available potassium (mg/kg).

**Table 2 Physicochemical properties of sands in the dried bottom of the Aral Sea.**

| Sand | SOC (g/kg) | SOM (g/kg) | TN (g/kg) | TP (g/kg) | TK (g/kg) | AN (mg/kg) | AP (mg/kg) | AK (mg/kg) | pH |
|---|---|---|---|---|---|---|---|---|---|
| Mean contents | 0.87 | 1.53 | 0.52 | 0.23 | 0.71 | 1.24 | 1.13 | 42.95 | 8.5 |

Note:
SOC, Soil organic carbon (g/kg); SOM, soil organic matter (g/kg); TN, total nitrogen (g/kg); TP, total phosphorous (g/kg); TK, total potassium (g/kg); AN, available nitrogen (mg/kg); AP, available phosphorous (mg/kg); AK, available potassium (mg/kg).

**Table 3 The salinity level of the sands in the dried bottom of the Aral Sea.**

| Sand | HCO3 (g/kg) | CI (g/kg) | SO$_4$ (g/kg) | Ca (g/kg) | Mg (g/kg) | Na (mg/kg) | K (mg/kg) |
|---|---|---|---|---|---|---|---|
| Mean contents | 0.02 | 4.24 | 12.43 | 2.12 | 1.48 | 632.22 | 43.24 |

investigated that biochar improves mungbean's physiological and anatomical traits under salt stress. Alfalfa, amaranth and maize were selected as plants that can adapt to the salty sand of the dry bottom of the Aral Sea. The effect of biochar on improve of the flora of the salty sand of the dry bottom of the Aral Sea was investigated. The present study evaluated the beneficial impact of biochar on growth and physiological characteristics in alfalfa, amaranth and maize and soil enzyme activities under saline sand conditions.

## MATERIALS AND METHODS

### Biochar, soil, and seed

The NUU Biology faculty's Soil Sciences department provided the municipal solid waste biochar. At 500 °C for 40 min, biochar made from municipal solid waste was pyrolyzed. Table 1 lists the qualities of the municipal solid waste biochar.

Sand samples were collected from the dry bottom of the southern part of the Aral Sea. Analysis of the sand's physicochemical properties and the salinity level of the sands in the dry bottom of the Aral Sea are shown in Tables 2 and 3.

Alfalfa (*Medicago sativa* L.), amaranth (*Amaranthus caudatus* L.) and maize (*Zea mays* L.) seeds were used for field experiments.

### Experimental design

The experimental work was conducted on saline sands in the dry bottom of the Aral Sea. The experiment was conducted to study the effect of biochar on the growth, root morphological and physiological traits of alfalfa, amaranth and maize. The Aral Sea's dry bottom served as the site for the experiment, which was conducted using a randomized block design with three replications. Experimental treatments included: T1—alfalfa (Control), T2—alfalfa (Biochar), T3—Amaranth (Control), T4—Amaranth (Biochar), T5

—Maize (Control), and T6—Maize (Biochar). After 40 days, plants were harvested and plant height, shoot and root fresh weights were measured.

## Measurement of root parameters

The roots of alfalfa, amaranth, and maize were cleaned with water with extreme caution. The entirety of the root system was dissected using a scanning system, and the results were evaluated with the help of the Win RHIZO program.

## Measurement of physiological traits

Photosynthetic pigments in alfalfa, amaranth and maize were measured spectrophotometrically by the method of *Hiscox & Israelstam (1979)*. Fresh leaves (50 mg) of alfalfa, amaranth and maize samples were cut and added dimethylsulfoxide (5 mL) to test tubes. The test tubes were incubated at 37 °C for 4 h. Then absorbance of extract was determined using a spectrophotometer. The relative water content of leaves in alfalfa, amaranth and maize was analyzed by the method of *Barrs & Weatherley (1962)*. Fresh leaf (100 mg) of ginger sample was placed in petri plates and added water in plates for 4 h. After 4 h, water content of leaves in alfalfa, amaranth and maize was measured.

## Analysis of soil nutrients

Following cultivation, the soil's agrochemical characteristics were evaluated. Using the improved procedure, the carbon (C) content and humus content (*GOST of Commonwealth of Independent States, 2003*). The P, K and total N contents in soil were analyzed by the method (*GOST 26261-84, 2005*, *GOST of Commonwealth of Independent States, 2002*).

## Analysis of soil enzymes

Urease activity of soil was determined using the method by *Pansu & Gautheyrou (2006)*. Soil samples (2.5 g) were added with toluene (0.5) mL for 15 min. After mixing, urea (2.5 mL) and citrate buffer were added to 5 mL. Incubator at 38 °C for 24 h. The urease activity was determined using a spectrophotometer. The soil enzyme activities (invertase and catalase) were assayed in the soil according to method by *Khaziev (2005)*.

## Statistical analyses

Experimental data were analyzed SPSS version 29 for Mac OS. Data were described in terms of mean and standard deviations. The influence of Biochar and different plants were assessed statistically using two-and one-way analysis of variance (ANOVA) and MANOVA. For further comparisons between groups Duncan's Multiple Range Test was applied at 0.05 level. Heatmap was generated using PAST statistical software.
The magnitude of the F-value determined the significance of the effect of treatment ($p < 0.05$, $<0.01$, and $<0.001$).

## RESULTS

The results in Table 4 show the effect of biochar on morphological traits of alfalfa, amaranth and maize under salt stress. Morphological traits of alfalfa, amaranth and maize were significantly decreased by salt stress.

**Table 4 Influence of biochar on growth of alfalfa, amaranth and maize in salt stress.**

| Plant species | Treatments | Plant height (cm) | Shoot fresh weight (g) | Root fresh weight (g) | Biomass allocation |
|---|---|---|---|---|---|
| Alfalfa | Control | $15.33 \pm 1.52^d$ | $0.04 \pm 0.01^e$ | $0.03 \pm 0.01^e$ | $1.46 \pm 0.33^a$ |
| | Biochar | $19.66 \pm 0.10^b$ | $0.06 \pm 0.02^c$ | $0.04 \pm 0.01^d$ | $1.41 \pm 0.03^a$ |
| Amaranth | Control | $13.67 \pm 1.53^e$ | $0.11 \pm 0.01^b$ | $0.07 \pm 0.01^b$ | $1.47 \pm 0.09^a$ |
| | Biochar | $16.93 \pm 0.90^{cd}$ | $0.17 \pm 0.01^a$ | $0.10 \pm 0.01^a$ | $1.73 \pm 0.02^a$ |
| Maize | Control | $18.00 \pm 1.00^c$ | $0.04 \pm 0.1^e$ | $0.04 \pm 0.01^d$ | $0.86 \pm 0.03^b$ |
| | Biochar | $26.60 \pm 0.53^a$ | $0.05 \pm 0.01^d$ | $0.06 \pm 0.01^c$ | $0.86 \pm 0.02^b$ |
| Repeated measure ANOVA | | | | | |
| Plant | | $<0.001^{***}$ | $<0.001^{***}$ | $<0.001^{***}$ | $<0.001^{***}$ |
| Treatment | | $<0.001^{***}$ | $<0.001^{***}$ | $<0.001^{***}$ | $0.556^{ns}$ |
| Plant × treatment | | $<0.001^{***}$ | $<0.001^{***}$ | $0.507^{ns}$ | $0.514^{ns}$ |

Notes:
\* significant at $p < 0.05$, \*\* significant at $p < 0.01$, \*\*\* significant at $p < 0.001$; ns, non-significant at $p > 0.05$ according to two way ANOVA.
[a,b] Means followed by different letters vertically (in the same column) are significantly different according to DMRTs.

**Table 5 Influence of biochar on root parameters of alfalfa, amaranth and maize under salt stress.**

| Treatments | | Total root length (cm) | Root projected area (cm²) | Root surface area (cm²) | Root volume (cm³) | Root diameter (mm) |
|---|---|---|---|---|---|---|
| Alfalfa | Control | $15.53 \pm 2.13^f$ | $1.80 \pm 0.03^f$ | $5.80 \pm 0.11^e$ | $0.11 \pm 0.01^e$ | $0.72 \pm 0.03^c$ |
| | Biochar | $22.68 \pm 0.79^e$ | $2.51 \pm 0.04^e$ | $7.60 \pm 0.02^e$ | $0.20 \pm 0.01^d$ | $1.06 \pm 0.03^a$ |
| Amaranth | Control | $38.72 \pm 2.00^d$ | $5.48 \pm 0.06^d$ | $16.73 \pm 0.50^d$ | $0.44 \pm 0.03^b$ | $0.89 \pm 0.03^b$ |
| | Biochar | $58.38 \pm 2.00^c$ | $6.86 \pm 0.07^c$ | $20.77 \pm 0.83^c$ | $0.60 \pm 0.02^a$ | $1.13 \pm 0.01^a$ |
| Maize | Control | $117.85 \pm 3.05^b$ | $9.76 \pm 0.51^b$ | $25.03 \pm 2.11^b$ | $0.33 \pm 0.02^c$ | $0.67 \pm 0.04^c$ |
| | Biochar | $269.51 \pm 2.08^a$ | $12.87 \pm 0.21^a$ | $40.90 \pm 0.26^a$ | $0.47 \pm 0.02^b$ | $0.96 \pm 0.02^b$ |
| Two-way ANOVA | | | | | | |
| Plant | | $<0.001^{***}$ | $<0.001^{***}$ | $<0.001^{***}$ | $<0.001^{***}$ | $<0.001^{***}$ |
| Treatment | | $<0.001^{***}$ | $<0.001^{***}$ | $<0.001^{***}$ | $<0.001^{***}$ | $<0.001^{***}$ |
| Plant × treatment | | $<0.001^{***}$ | $<0.001^{***}$ | $<0.001^{***}$ | $0.011^*$ | $0.274^{ns}$ |

Notes:
\* significant at $p < 0.05$, \*\* significant at $p < 0.01$, \*\*\* significant at $p < 0.001$; ns, non-significant at $p > 0.05$ according to two way ANOVA.
[a,b] Means followed by different letters vertically (in the same column) are significantly different according to DMRTs.

Alfalfa (biochar) treatment significantly increases the plant height by 28%, shoots fresh weight by 50% and roots' fresh weight by 33% as compared to control under salt stress. In salt conditions, amaranth (biochar) treatment significantly enhanced the plant height by 24%, shoot fresh weight by 54% and root fresh weight by 43% than the control. The additions of biochar to the sand caused significantly enhanced the plant height by 48%, shoot fresh weight by 25% and root fresh weight by 50% in maize under salt stress.

Data revealed that biochar treatments significantly improved root parameters as well as salt stress control (Table 5.). Compared to the control, alfalfa (biochar) treatment significantly increased the root projected area by 39% and the root surface area by 31% under salt stress. The total root length, diameter and volume were sharply enhanced by alfalfa (biochar) treatment, which significantly increased by 46%, 47% and 82%, respectively, to the control. The root surface area, projected area, and diameter were improved with the amaranth (biochar) by 24%, 25% and 27% to the control in salt stress.

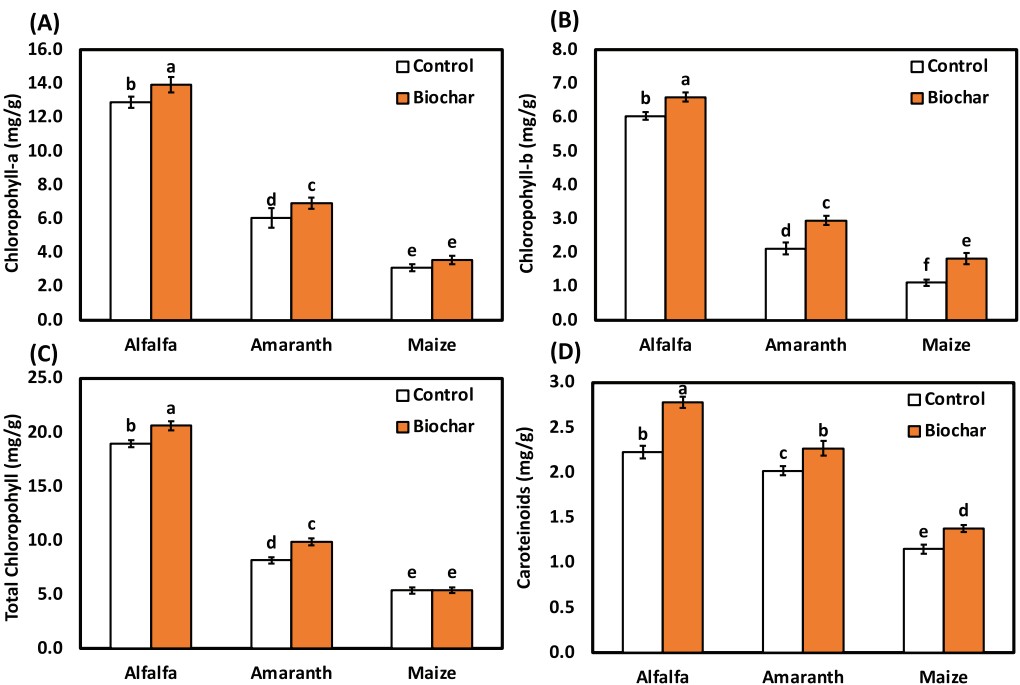

**Figure 1 Influence of biochar on photosynthetic pigments of alfalfa, amaranth and maize under salt stress.** Bars followed by different letters are significantly different according to DMRTs at 0.05 level.

In salt stress, amaranth (biochar) treatment significantly increased the root volume by 36% and the total root length by 50%. The highest values of root surface area (63%) were observed in the treatment of maize (biochar) than in the control. In salt stress conditions, maize (biochar) significantly increases the total root length by 24%, root projected area by 32%, root volume by 42% and root diameter by 43% more than the control.

Data revealed that biochar treatments increased photosynthetic pigments of alfalfa, amaranth and maize as compared with the control under salt stress (Fig. 1). Alfalfa (biochar) treatment significantly enhanced the chlorophyll a content by 8%, chlorophyll b content by 10%, total chlorophyll content by 9% and carotenoid content by 25% than the control under salt stress. Under salt stress, amaranth (biochar) treatment significantly enhanced chlorophyll b content by 39%, total chlorophyll content by 21% and carotenoid content by 12% than the control. The highest level of chlorophyll b content was observed in maize (biochar) treatment recording a significant increase of 64% over the control under salt stress.

Data in Fig. 2 indicated that salinity decreased the relative water content of leaf in alfalfa, amaranth and maize. Compared to the control, all biochar treatments improved the relative water content of leaf in alfalfa, amaranth and maize. However, amaranth's maximum relative water content was detected in amaranth (biochar) treatment, respectively. The results in Table 6 show the effective impact of biochar on soil nutrients under salt stress.

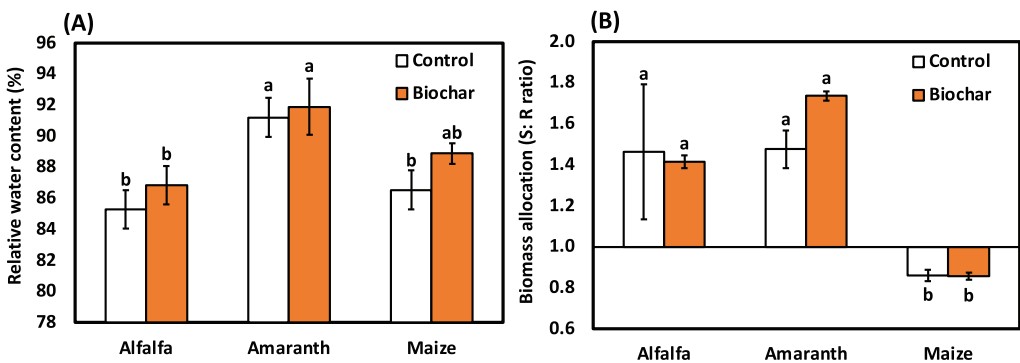

**Figure 2** Influence of biochar on (A) The relative water content and (B) Biomass allocation (shoot: root ratio) of alfalfa, amaranth and maize under salt stress. Bars followed by different letters are significantly different according to DMRTs.

**Table 6 Influence of biochar on soil nutrients under salt stress.**

| Treatments | | N (mg/kg) | P (mg/kg) | K (mg/kg) | Humus (%) |
|---|---|---|---|---|---|
| Alfalfa | Control | 4.00 ± 0.02[b] | 4.15 ± 0.01[b] | 50.45 ± 0.26[d] | 0.09 ± 0.01[e] |
| | Biochar | 4.71 ± 0.02[a] | 4.82 ± 0.05[a] | 52.13 ± 0.65[c] | 0.14 ± 0.01[c] |
| Amaranth | Control | 3.05 ± 0.01[d] | 2.13 ± 0.05[e] | 55.1 ± 1.46[b] | 0.10 ± 0.01[e] |
| | Biochar | 3.57 ± 0.01[c] | 2.45 ± 0.09[d] | 64.7 ± 0.35[a] | 0.12 ± 0.01[d] |
| Maize | Control | 2.92 ± 0.01[d] | 2.35 ± 0.01[d] | 51.6 ± 2.01[cd] | 0.18 ± 0.01[b] |
| | Biochar | 3.01 ± 0.06[d] | 2.92 ± 0.05[c] | 64.9 ± 0.86[a] | 0.23 ± 0.01[a] |

Notes:
[*] significant at $p < 0.05$, [**] significant at $p < 0.01$, [***] significant at $p < 0.001$; ns, non-significant at $p > 0.05$ according to two way ANOVA.
[a,b] Means followed by different letters vertically (in the same column) are significantly different according to DMRTs.

The soil nutrients of N, P, K and humus contents were promoted by applying biochar treatments. The alfalfa (biochar) treatment significantly increased N content by 18%, P content by 16%, K content by 17% and humus by 55%, respectively, than the control. The P, N, K and humus contents increased by 15%, 17%, 17% and 20% when inoculated with amaranth (biochar) treatment. However, the highest P and K contents were observed with maize (biochar) treatment. Maize (biochar) treatment had a greater effect on increasing P content (24%), K content (26%) and humus content (28%).

The impact of biochar treatments on enzyme activities in salt stress is given in Table 7. Alfalfa (biochar) treatment significantly enhanced the catalase by 30%, invertase by 40%, and urease by 25% in soil under saline conditions. The catalase and urease activities were improved by 12% and 17%, respectively, when the soil was amended by amaranth (biochar) over the control in salt stress. Under salt stress, the catalase and urease activities increased by 13% and 24% in maize (biochar) treatment.

The overall impact of biochar treatments on plant growth, physiology and enzyme activities under salt stress is given in Table 8 in terms of multivariate analysis of variance presenting the impact of biochar treatment, different plants, and the interaction between plants and biochar. Alfalfa (biochar). An overall highly significant effect of biochar was recognized in most of the studied variables, including; Plant height, shoot fresh weight

**Table 7 Influence of biochar on enzyme activities of soil under salt stress.**

| Treatments | | Catalase (mL $KMnO_4$ $g^{-1}$ soil $h^{-1}$) | Invertase (µg glucose·$g^{-1}$ soil·$h^{-1}$) | Urease ($NH_4$/g of soil/h) |
|---|---|---|---|---|
| Alfalfa | Control | $2.18 \pm 0.02^f$ | $1.52 \pm 0.01^e$ | $1.42 \pm 0.01^b$ |
| | Biochar | $2.83 \pm 0.04^e$ | $2.13 \pm 0.01^d$ | $1.77 \pm 0.01^a$ |
| Amaranth | Control | $7.00 \pm 0.05^b$ | $5.62 \pm 0.03^b$ | $1.23 \pm 0.01^c$ |
| | Biochar | $7.81 \pm 0.02^a$ | $6.04 \pm 0.02^a$ | $1.44 \pm 0.01^b$ |
| Maize | Control | $3.98 \pm 0.02^d$ | $3.25 \pm 0.01^c$ | $1.12 \pm 0.01^d$ |
| | Biochar | $4.49 \pm 0.02^c$ | $3.55 \pm 0.01^c$ | $1.39 \pm 0.01^b$ |

Notes:
[*] significant at $p < 0.05$, [**] significant at $p < 0.01$, [***] significant at $p < 0.001$; ns, non-significant at $p > 0.05$ according to two way ANOVA.
[a,b] Means followed by different letters vertically (in the same column) are significantly different according to DMRTs.

**Table 8 Multivariate analysis of variance showing the influence of biochar treatments on different plants physiological parameters.**

| Dependent variable | Corrected model | | Plant | | Treatment | | Plant * treatment | |
|---|---|---|---|---|---|---|---|---|
| | F | p-value | F | p-value | F | p-value | F | p-value |
| Plant height | 71.0 | <0.001*** | 88.3 | <0.001*** | 151.0 | <0.001*** | 13.7 | <0.001*** |
| SFW | 920.5 | <0.001*** | 1,937.2 | <0.001*** | 548.6 | <0.001*** | 89.8 | <0.001*** |
| RFW | 68.3 | <0.001*** | 140.7 | <0.001*** | 58.5 | <0.001*** | 0.7 | 0.507ns |
| Biomass allocation | 6.6 | 0.004** | 15.6 | <0.001*** | 0.4 | 0.556ns | 0.7 | 0.514ns |
| Total length | 6,273.0 | <0.001*** | 11,743.5 | <0.001*** | 3,568.6 | <0.001*** | 2154.7 | <0.001*** |
| Total Proj area | 565.2 | <0.001*** | 1,318.8 | <0.001*** | 140.6 | <0.001*** | 23.8 | <0.001*** |
| Tot surf area | 483.4 | <0.001*** | 1,010.4 | <0.001*** | 229.5 | <0.001*** | 83.5 | <0.001*** |
| Volume | 327.3 | <0.001*** | 686.8 | <0.001*** | 249.2 | <0.001*** | 6.8 | 0.011* |
| Diameter | 46.0 | <0.001*** | 26.1 | <0.001*** | 175.2 | <0.001*** | 1.4 | 0.274ns |
| RWC | 4.3 | 0.018* | 9.5 | 0.003** | 2.1 | 0.171ns | 0.2 | 0.813ns |
| Chl-a | 455.4 | <0.001*** | 1,127.7 | <0.001*** | 19.9 | <0.001*** | 0.9 | 0.439ns |
| Chl-b | 824.0 | <0.001*** | 2,002.2 | <0.001*** | 113.1 | <0.001*** | 1.4 | 0.282ns |
| Total chlorophyll | 1,284.3 | <0.001*** | 3,170.3 | <0.001*** | 54.3 | <0.001*** | 13.3 | <0.001*** |
| Carotenoid | 307.5 | <0.001*** | 681.4 | <0.001*** | 148.0 | <0.001*** | 13.5 | <0.001*** |
| Catalase | 594.2 | <0.001*** | 1,450.2 | <0.001*** | 69.3 | <0.001*** | 0.8 | 0.483ns |
| Invertase | 229.8 | <0.001*** | 563.4 | <0.001*** | 20.8 | <0.001*** | 0.9 | 0.441ns |
| Urease | 60.2 | <0.001*** | 38.0 | <0.001*** | 178.8 | <0.001*** | 23.0 | <0.001*** |
| N | 53.1 | <0.001*** | 105.4 | <0.001*** | 54.8 | <0.001*** | 0.0 | >0.999ns |
| P | 806.5 | <0.001*** | 1,917.1 | <0.001*** | 174.7 | <0.001*** | 11.8 | 0.001*** |
| K | 243.2 | <0.001*** | 315.2 | <0.001*** | 455.0 | <0.001*** | 65.3 | <0.001*** |
| Humus | 26.6 | <0.001*** | 9.1 | 0.004** | 96.6 | <0.001*** | 9.1 | 0.004** |

Note:
[*] significant at $p < 0.05$, [**] significant at $p < 0.01$, [***] significant at $p < 0.001$; ns, non-significant at $p > 0.05$ according to MANOVA.

(SFW), root fresh weight (RFW), total length, total root ProjArea, total surface area, volume, diameter, Chl-a, Chl-b, total chlorophyll, carotenoid, catalase, invertase, Urease, N, P, K and Humus. However, biomass allocation and relative water contents showed non-significant impact of biochar treatment. The difference on three plant species was

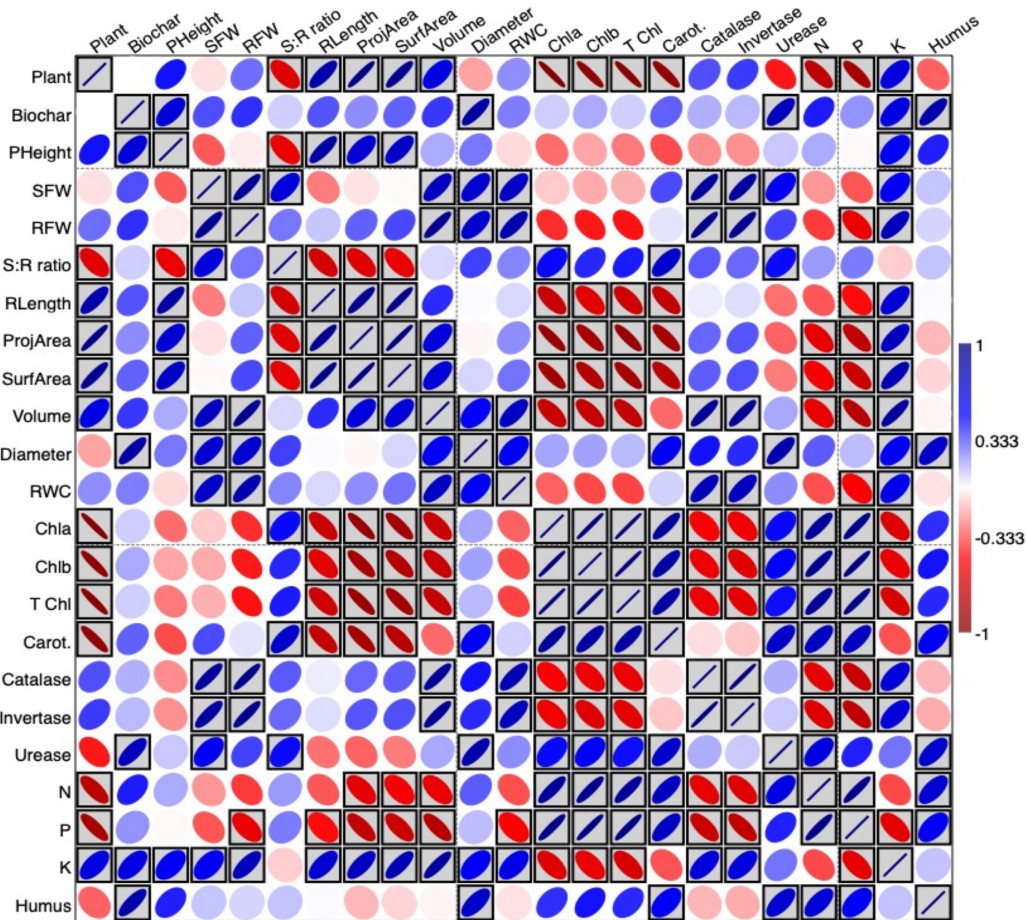

**Figure 3 Heatmap presenting the interrelationship between study variables in terms of Pearson's correlation test.**

significant in all studied parameters as revealed by MANOVA. The interaction between biochar treatment and plant species was significant in plant height, shoot fresh weight, total root length, total root proj. area, root surface area, root volume, total chlorophyll, carotenoids, urease activities, P, K, and humus.

The interrelationship between study variables in terms of Pearson's correlation test was plotted as a heatmap presented in Fig. 3 using PAST software. The blue boxes indicate a positive correlation between variables; the red boxes indicate a negative correlation and the grey boxed colors for significant correlations. Accordingly, biochar treatments had a positive correlation with most of the growth parameters measured, and significantly in plant height, diameter, urease, K and humus (%). Moreover, N and P were significantly and positively correlated to pigment concentrations.

## DISCUSSION

Salt stress is one of the focal contrary environmental factors that restrict plant growth in the world. It can cause a marked reduction in plant height, root length, plant fresh weight, dry weight and the productivity of many crops (*Huang et al., 2017*; *Ibrahim et al., 2020*,

*2021*; *Kumar et al., 2021b*). In this present study, the chlorophyll a content, chlorophyll b content, total chlorophyll content and carotenoid content in leaves of maize, amaranth and alfalfa was also affected by salinity (Fig. 1). Moreover, chlorophyll a and total chlorophyll contents of tomato plants significantly decreased (*Kul et al., 2021*). The active photosynthetic process in leaves was also damaged, leading to chlorosis and early leaf senescence under salinity conations. Similarly, *Zahra, Raza & Mahmood (2020)* observed a reduction in chlorophyll content and carotenoid content in maize genotypes under salt stress. Photosynthetic pigments and plant nutrients in winter and spring wheat showed a sharp decrease under salt stress (*Saddiq et al., 2021*). Also, numerous researchers have reported that biochar increased nutrient uptake (*Ndiate et al., 2021*) and plant physiological properties (*Kul et al., 2021*; *Zhang et al., 2020*) in various plants under salinity stress.

In the current study, biochar improved photosynthetic pigments such as the total chlorophyll content and carotenoid content in alfalfa, amaranth and maize. Integrating biochar in the current study could enhance the morphological characteristics of the chosen plants under salinity stress. This finding is consistent with the report of *Hussein et al. (2022)*, who observed biochar improved the root length and plant growth in spinach (*Spinacia oleracea* L.) under salt stress. The prospective benefits of biochar on leaf area index, photosynthetic potential, transpiration rates and chlorophyll content significantly helped the leaf photosynthesis and net assimilation rate of rice growth and production (*Huang et al., 2022*; *Piao et al., 2023*). Applying biochar greatly improved the salt tolerance of cabbage seedlings and considerably increased chlorophyll a, b and total chlorophyll while lowering sucrose, proline and $H_2O_2$. It was revealed by *Alfadil et al. (2021)* that the addition of biochar under salinity stress had a substantial impact on maize's ability to absorb nutrients and maintain water status, and biochar could reduce osmotic stress through increasing soil water content and releasing mineral nutrients in the soil solution and plants by its high transient sodium ion binding due to its high adsorption ability. Biochar could attract additional $Na^+$ into the soil under salinity conditions, releasing nutrients and lowering osmotic stress by increasing water holding capacity and $CO_2$ absorption, which ultimately led to a noticeable improvement in photosynthetic, stomatal conductance and transcription rates (*Ibrahim et al., 2021*). Additionally, biochar was able to reduce $Na^+/K^+$ ratio and the amount of Na in many plants, which lessened the negative impact of salts on plants (*Ali et al., 2017*; *Huang et al., 2022*; *Lashari et al., 2015*).

Biochar application could directly improve the nitrogen status of salt-affected soils. Also, it indirectly impacted the quantity and activity of bacteria, which could drive N transformation and enhance nitrogen release (*Yao et al., 2021*). Moreover, a considerable rose in soil nitrogen was recorded, and this increase was closely correlated with the applied dose of biochar. These findings confirmed the ability of biochar to hold the nutrients. The amount of soil organic matter was dramatically increased due to biochar application. In severely saline-sodic soil, marked increases in soil organic matter and nutrients after biochar addition. Numerous studies suggested that biochar could directly act as a P source and indirectly improve soil texture to boost P status in salt-affected soils (*Saifullah et al., 2018*). Similar findings were also informed by *Alfadil et al. (2021)* biochar increased

nutrient contents in the soil under salt stress. *Premalatha, Malarvizhi & Parameswari (2022)* reported that biochar increases soil organic C and available N, P and K under salt stress. Similarly, *Yao et al. (2021)* have reported that biochar promoted the availability of soil total N, P and K under saline conditions. Similar results have been informed by *Huang et al. (2022)*. The biochar improved soil organic matter, total N, and P contents under salt stress. *Gunarathne et al. (2020)* reported that biochar promoted soil's catalase and alkaline phosphatase activity in salt stress conditions.

Soil microbial activity has been exemplified by enzyme activity, which is susceptible to changes in the soil environment (*Du, 2019*; *Elzobair et al., 2016*; *Huang et al., 2017*; *Khadem & Raiesi, 2017*). In addition, soil enzyme activities were critical for soil organic matter degradation and nutrient cycling (*Yao et al., 2021*) and are considered important indicators of soil quality. However, few studies focused on biochar's effect on enzyme activities in saline soils. Our results showed that adding biochar positively affected enzyme activity in saline soils. The conversion of soil organic nitrogen into usable inorganic nitrogen was related to urease. Invertase was crucial in increasing the amount of soluble nutrients in the soil, which gave soil organisms enough energy. Catalase was a major factor in the oxidation of organic matter and humus production and it of the level of biological redox and microbial activity in the soil. Compared to control treatments, biochar application had a significant effect on the activities of catalase, invertase, and urease in the current investigation (*Yao et al., 2021*). The use of biochar could enhance enzyme activity by enhancing SOM, microbial activity, and microbial biomass or by placing the enzymes close together and allowing them to interact with the biochar surface (*Azadi & Raiesi, 2021*; *He et al., 2020*; *Qu et al., 2020*).

## CONCLUSION

Salinity stress had a negative effect on growth, root morphological traits and photosynthetic pigments of alfalfa, amaranth and maize. The amount of chlorophyll a and b, and carotenoid contents in of alfalfa, amaranth and maize decreased by salinity. Biochar treatments have more stimulation impact on most of the root morphological traits and plant growth parameters of alfalfa, amaranth and maize compared to the control in salt stress. In salt stress, biochar application could improve photosynthetic pigments viz: the total chlorophyll content, chlorophyll a and chlorophyll b. The application of biochar treatments positively improved nutrients of N, P, K and humus contents in the soil in salt stress. The stimulation impact of biochar treatments on increasing soil enzyme activities such as catalase, invertase and urease and enhancing nutrient availability can reduce the application of the chemical fertilizers in salt stress.

### Funding

The authors received no funding for this work.

## Competing Interests

Amr Elkelish is an Academic Editor for PeerJ.

## Author Contributions

- Dilfuza Jabborova conceived and designed the experiments, analyzed the data, authored or reviewed drafts of the article, and approved the final draft.
- Tokhtasin Abdrakhmanov conceived and designed the experiments, performed the experiments, analyzed the data, prepared figures and/or tables, authored or reviewed drafts of the article, and approved the final draft.
- Zafarjon Jabbarov conceived and designed the experiments, performed the experiments, analyzed the data, prepared figures and/or tables, authored or reviewed drafts of the article, and approved the final draft.
- Shokhrukh Abdullaev conceived and designed the experiments, analyzed the data, prepared figures and/or tables, authored or reviewed drafts of the article, and approved the final draft.
- Abdulahat Azimov conceived and designed the experiments, prepared figures and/or tables, authored or reviewed drafts of the article, and approved the final draft.
- Ibrahim Mohamed performed the experiments, analyzed the data, prepared figures and/or tables, authored or reviewed drafts of the article, and approved the final draft.
- Maha AlHarbi performed the experiments, analyzed the data, prepared figures and/or tables, authored or reviewed drafts of the article, and approved the final draft.
- Abdelghafar Abu-Elsaoud performed the experiments, analyzed the data, prepared figures and/or tables, authored or reviewed drafts of the article, and approved the final draft.
- Amr Elkelish conceived and designed the experiments, analyzed the data, prepared figures and/or tables, authored or reviewed drafts of the article, and approved the final draft.

## Data Availability

The raw data are available in the Supplemental File.

## Supplemental Information

Supplemental information for this article can be found online at http://dx.doi.org/10.7717/peerj.15684#supplemental-information.

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
