# Peer review of "Biochar improves the growth and physiological traits of alfalfa, amaranth and maize grown under salt stress"

_PeerJ, doi:10.7717/peerj.15684_

## Round 0.1 · original submission · Minor Revisions

Dear Authors

According to the reviewer's comments, the manuscript needs a minor revision to be reconsidered for publication. The authors are invited to revise the paper considering all the suggestions made by the reviewers. Please note that requested changes are required for publication.

Best Regards

·

Basic reporting

The manuscript is well written but still needs some minor changes

Experimental design

No comments

Validity of the findings

No comments

Additional comments

I reviewed the paper titled "Biochar ImprovestheGrowth and Physiological Traits of Alfa-alfa, Amaranth and Maize Grown under Salt Stress". The authors aimed to evaluate the beneficial impact of biochar on alfalfa, amaranth and maize growth and physiological characteristics under salt stress conditions. The results are interesting and applicable. The manuscript needs minor changes.
-Comments and Suggestions for Authors
Abstract
-More details are needed in the method part
- Please find more corrections as track changes in the manuscript PDF file.
Introduction
-The introduction section is comprehensive and well written.
-Please find more corrections as track changes in the manuscript PDF file.
Materials and methods
-This section needs more details to enable the other researchers to repeat any experiment
-Please find more corrections as track changes in the manuscript PDF file.
Results
-The results section is well written. I suggest the authors to unify the manner of presenting results either using tables or figures. For example Tables 4, 5, 6 qnd 7 which show the Influence of biochar on growth of alfalfa, amaranth and maize in salt stress,……..etc, and Figure 1 which show the Influence of biochar on photosynthetic pigments of alfalfa, amaranth and maize under salt stress, and Figure 2 which show the Influence of biochar on (A) the relative water content and (B) biomass allocation (shoot:root ratio) of alfalfa, amaranth and maize under salt stress.
-The results section is well written. I suggest the authors to use error bars with standard error to decrease the bar length in all figures
- Please find more corrections as track changes in the manuscript PDF file.
Discussion
-The discussion section is well written.
Conclusion
-The conclusion section is well written.
References
Please unify the style according to the journal instructions

·

Basic reporting

1. The authors targeted a good topic, but the article seems to have very poor presentation. The English writing is very poor throughout the manuscript, and there are many typos.
2. The abstract and introduction are poorly written. The objectives were not clear.

Experimental design

3. Methodology writing is not sufficient. Why this species were used need to clarify. Besides, the methods and concentration of salt and BC have not been properly mentioned.
4. The objectives are not clear. because the authors used soil properties data rather than plant physiological properties under salt and BC application

Validity of the findings

5. Data presentation should be improved. Plant morphological and physiological data should be considered under salt stress and BC application rather than soil data.
6. The conclusion is unclear because the majority of the experimental data is from soil studies, but they mention BC alleviates salt stress in plants. Its strange.
7. The article is serious lack of presentation and data collection.

Reviewer 3 ·

Basic reporting

The manuscript entitled" Biochar improves the growth and physiological traits of Alfaalfa, Amaranth and Maize grown under salt stress" . the paper was written in scientific method with very minor english mistakes. I recommend the publishing of this paper .
some observation will be discussed
1- introduction
can you add paragraph about the role of different enzymes related to salt stress (SOD, POD, APX and so one.
2- the role of the studied enzymes (catalse, urease and invertase) with expression of different specific genes that overcome the salinity stress.
3- Discussion : need to be more clarify

Experimental design

good

Validity of the findings

good

Reviewer 4 ·

Basic reporting

Literature references, sufficient field background/context provided

Experimental design

Rigorous investigation performed to a high technical & ethical standard

Validity of the findings

Conclusions are well stated, linked to original research question & limited to supporting results

Additional comments

The topic of this paper is well-chosen, useful and innovative. However, it needs to be improved and thoroughly revised before it can be accepted for publication in the Peer J. Greater attention should be paid to the following points:
1. The purpose of the research work is not sufficiently explained. It has not been specified why alfalfa, amaranth and maize were selected as research material.
2 In the introduction part physicochemical properties of soil are more emphasized compared to the physiological properties of plants under salinity.
3 Line 174-175 - It was written that morphological traits decreased under salinity. However, there is no data for morphological traits under 0 mM NaCl to compare.
4 Line 224 - There is no explanation for abbreviations-SFW, RFW
5 Line 243 - There is no reference. It will be better to include the results of previous works related to chlorophyll content under salinity of more plants not only a tomato.
6 Line 268- Na/K ratio should be changed into Na+/K+
7 Results of previous work for maize, alfalfa and amaranth were discussed little in the discussion part also.
8 There is no explanation for abbreviations in Table 1 and Table 2
9 References should be double-checked and formatting should be done strictly according to journal rules.

---

## Round 0.2 · accepted · Accept

Dear Authors

I am pleased to inform you that after the last round of revision, the manuscript has been improved a lot, and it can be accepted for publication.

Congratulations on accepting your manuscript, and thank you for your interest in submitting your work to PeerJ.

With Thanks

The Section Editor noted:

> Authors are inconsistent in species name for alfalfa, sometimes using "alfalfa", sometimes using "alfa-alfa". Capitalization is also not consistent for alfa-alfa. Please have them revise to be consistent ("alfalfa" preferred). line 35 typo "sat" --> "salt"

·

Basic reporting

No comment

Experimental design

No comment

Validity of the findings

No comment

Additional comments

The manuscript has been improved accordingly. My recommendation is the article should be accepted as is.

Reviewer 3 ·

Basic reporting

The revised manuscript was written in good format and the comments of the reviweres.

Experimental design

good

Validity of the findings

good

Additional comments

Paper was better than first submitting